# Lifelong Autonomous Fine-Tuning of Navigation Foundation Models in the Wild

**Kyle Stachowicz**[*], **Lydia Ignatova**[*], **Sergey Levine**
UC Berkeley
{kstachowicz, lydiaignatova, svlevine}@berkeley.edu

**Abstract:** Recent works have proposed a number of general-purpose robotic foundation models that can control a variety of robotic platforms to perform a range of different tasks, including in the domains of navigation and manipulation. However, such models are typically trained via imitation learning, which precludes the ability to adapt autonomously through experience that the robot gathers on the job. In this work, our aim is to train general-purpose robotic foundation models in the domain of robotic navigation specifically with the aim of enabling autonomous self-improvement. We show that a combination of pretraining with offline reinforcement learning and a complete system for continual autonomous operation leads to a robotic learning framework that not only starts off with broad and diverse capabilities, but can further specialize and adapt those capabilities in the course of carrying out navigational tasks in a given deployment location. To our knowledge, this result demonstrates the first navigation robot foundation model to continually learn via autonomous interaction in open-world settings.

**Keywords:** Reinforcement Learning, Robot Foundation Model

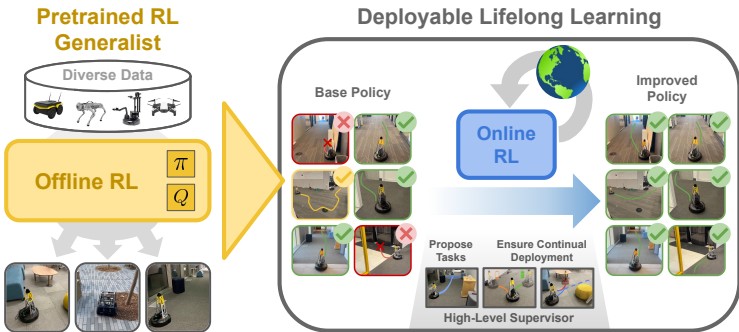

**Figure 1:** LiReN is a navigation foundation model that improves autonomously during deployment via reinforcement learning. Offline RL pretraining provides broad generalization capabilities, while lifelong fine-tuning allows the model to get better and better through deployment.

## 1 Introduction

Many recent developments in machine learning have been driven by the advent of large-scale foundation models that leverage pretraining on large-scale Internet data. The same general recipe – pretraining on diverse datasets and then adapting to new tasks via finetuning or in zero-shot – has delivered impressive results on a wide variety of tasks in natural language processing [1, 2], computer vision [3, 4], and other domains [5]. Analogously, *robot foundation models* – defined broadly as policies trained on diverse robot data that can be further fine-tuned for downstream applications – have shown promising results by training behavioral cloning policies on diverse multi-embodiment robot datasets [6, 7]. The process of fine-tuning these models typically requires training on expensive human demonstrations in the target environment.

---

[*] Equal contribution. Videos, code, and models: `kylestach.github.io/lifelong-rl-nav`

8th Conference on Robot Learning (CoRL 2024), Munich, Germany.

This paradigm cannot deliver models that improve or adapt without human data – for example, to respond to changes in the target environment or wear and tear on the robot itself. Ideally, a robot foundation model could instead continually improve via training on *autonomous* data collected while deployed. Such a model could adapt on the job, becoming better and better the more it is used without relying on human supervision. However, this requires a learning objective capable of learning from suboptimal data. While reinforcement learning is a natural formulation for this problem, it is nontrivial to directly fine-tune BC policies with RL. The sample-efficient RL algorithms that are most practical for use in the real world [8, 9] require learning a critic $Q(s, a)$, which requires different feature-learning capabilities than BC [10].

In this paper we ask: **how can we enable practical autonomous fine-tuning of generalist navigation policies in the real world?** We posit that continually improving generalist policies in the real world requires two main ingredients: a scalable training objective that is able to successfully incorporate suboptimal data, and a supervisor framework for proposing tasks, computing rewards, and managing real-world deployment concerns at a level of abstraction above the policy itself.

We instantiate these components in the visual navigation setting with our approach, **Li**felong **Re**inforcement learning for **N**avigation (LiReN, Fig. 1). Like prior work in foundation models for visual navigation, LiReN performs pretraining on a large, cross-embodiment visual navigation dataset. However, unlike existing methods, LiReN is trained with offline reinforcement learning to enable further training online. We wrap our foundation model with a framework to enable lifelong learning and show that LiReN is capable of autonomously improving its performance through online interactions over long timescales, increasing goal-reaching performance in the downstream environment from 40% to 75% without additional human data.

## 2    Related Work

**Mobile robots and visual navigation.**    We study the visual navigation problem, where a mobile robot navigates using only egocentric image observations to reach a goal pose, also specified by an egocentric image [11]. The task must be done in minimal time without colliding with obstacles. Previous work applied BC to visual navigation in both simulation [12, 13] and real-world environments [14]. Also related to our deployment setting is the CoBots system [15, 16, 17], a mobile robot navigation system successfully deployed in real environments, although it does not include learning components or online improvement. However, to our knowledge, LiReN is the first generalist navigation policy that can be continually improved during deployment with online RL.

**Robot foundation models.**    Several recent works have proposed *robot foundation models*, defined in this work as models trained on large-scale, diverse robot data (typically aggregated from many embodiments). The RT- series of models propose training generalist robot manipulation models via BC on single-embodiment (RT-1 [18], RT-2 [19]) and multi-embodiment (RT-X [20]) datasets. Similarly, Octo [6] proposes an open-source BC model for manipulation based on a scalable diffusion-based architecture. ViNT [7], the most relevant such model, trains a generalist policy on a broad navigation dataset and deploys on several mobile robot embodiments. While BC enables a simple learning objective, fine-tuning these models is challenging: further improvement requires expert trajectories, and fine-tuning BC policies with online RL is difficult without reward-specific features [10]. Pre-Training for Robotics [21] applies offline RL to diverse data and fine-tunes (also with offline RL) on demonstration data. While we similarly pre-train with offline RL, we also include an *autonomous* phase in which we fine-tune our offline foundation model with new online data.

**Online reinforcement learning.**    Another approach to learning an end-to-end robot model starts from scratch, learning a policy using only online RL. While traditional wisdom suggests training in sim for data efficiency reasons [22], recent work has enabled sample-efficient RL operating directly in the real world [9, 23]. Zhu et al. [24] further studies how to make such an online learning problem autonomous. The closest navigation work, FastRLAP [10], learns a high-speed navigation policy using an actor-critic algorithm applied to features from a RL-pretrained encoder. While this provides

a strong prior in the form of relevant features, the policy is not initialized with any notion of collision avoidance, goal-reaching, or other navigation behaviors. We instead train the entire policy and critic on offline data, providing a strong behavioral prior upon which the RL algorithm can improve.

**Autonomous improvement for robotics.** RoboCat [25] achieves autonomous improvement in the manipulation setting by collecting thousands of policy rollouts and then re-training the model from scratch on this dataset, filtered for successes. Isele et al. [26] learns low-level controllers for mobile robots in simulation. In contrast to these works – which consider a closed-world, instrumented environment in order to enable autonomous learning – we study RL-based autonomous fine-tuning in a real office building in the navigation setting, which is inherently open-world. We thus develop a supervisory framework to enable autonomous learning for our online fine-tuning system.

## 3 Preliminaries

We begin by formalizing our problem statement and discussing several prior works upon which our proposed framework builds:

**Visual navigation.** We formalize the visual navigation problem as a goal-conditioned partially-observed Markov decision process in which observations are given by egocentric images from the robot's camera and actions correspond to linear and angular velocity commanded to the robot, or the robot's relative position $K$ frames ahead for the offline data in which raw actions are not available [27]. We specify goals via a future observation from a pose we would like the robot to reach, and consider a trajectory as a success when the robot is within a threshold $\epsilon$ of this pose.

**Sample-efficient RL.** To learn from sub-optimal data collected autonomously in the real world, we turn to tools from reinforcement learning. We express our problem as a (goal-conditioned, partially observed) Markov decision process $\mathcal{M}$ composed of observation and action spaces $\mathcal{O}, \mathcal{A}$ and goals $\mathcal{G}$. We want to learn a policy $\pi : \mathcal{S} \times \mathcal{G} \rightarrow \mathcal{A}$ through interactions with the environment. In visual navigation tasks we define the observation space as a single image frame of input and consider goal-reaching tasks considered on observations taken from the desired state.

We use actor-critic methods for their ability to learn from off-policy data. These methods learn a $Q$-function corresponding to the expected sum of discounted returns starting in a state-action pair, and a policy $\pi$. In deep RL, $Q$ and $\pi$ are approximated with a neural network. The policy $\pi$ is trained to maximize $Q(s, a)$, which is learned via approximate dynamic programming:

$$y \leftarrow \mathbb{E}_{s' \sim \mathcal{M}, a' \sim \pi(\cdot|s')} \left[ r_t + Q(s', a') \right]; \quad \min \mathcal{L}_Q = \mathbb{E} \left[ (Q(s, a) - y)^2 \right]. \tag{1}$$

Actor-critic methods require continuous online interaction to avoid catastrophic overestimation [28]. When the policy is held stationary or very slow-moving compared to data collection this can be viable, but sample-efficiency typically increases with a high *update-to-data* (UTD) ratio between gradient steps and environment steps. Stabilizing high-UTD RL requires regularization techniques such as clipped double-$Q$ [29], ensembling [8], entropy regularization [30], and critic normalization [31]. In LiReN's implementation, we use a combination of all four of these techniques (following e.g. [10]), as well as several other techniques described in Sec. 4 and App. A. We consider a special case (similar in spirit to hindsight relabeling [32]) such that some additional information about $s_t$ becomes knowable *later* and provide this privileged information to the critic (but not the actor).

**Offline RL with conservative Q-learning.** To provide a high-quality initialization for our off-policy actor-critic algorithms, we require a pre-training method that is capable of learning both a policy $\pi$ and a critic function $Q$. One such choice of pretraining method is offline reinforcement learning, which attempts to solve the same maximum-expected-reward problem as online RL but with a static dataset rather than through online interaction. This requires even further regularization to ensure the policy's actions remain in the approximate support of the original dataset.

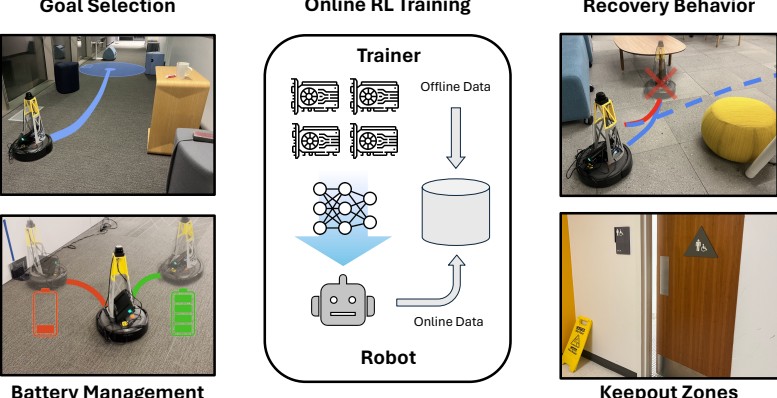

**Figure 2: Requirements for autonomous improvement.** Autonomous improvement mechanisms for proposing and evaluating goals, providing resets, and enabling safe operation. In our setting, pseudo-resets are enabled by randomized perturbations after failures, and safe behavior corresponds to avoiding designated zones and maintaining battery charge.

In particular, we apply conservative $Q$-learning [33], which proposes a conservative regularizer to artificially depress the $Q$-values of out-of-distribution actions compared to dataset actions. CQL adds the following term (blue) to the standard temporal-difference based $Q$-learning objective:

$$\min_\theta \mathbb{E}_{s,a\sim\mathcal{D}} \left[ (Q_\theta(s,a) - y)^2 + \mathbb{E}_{a'\sim\pi} \left[ Q(s,a') \right] - Q(s,a) \right]. \tag{2}$$

**Autonomously improving robot foundation models.**  Training reinforcement learning algorithms in the wild presents significant challenges not present in simulation. In the standard reinforcement learning framework, we assume that we have access to episodic **resets** which occur at regular intervals and ensure adequate state coverage, and each episode we are given a plausible **task**. Task success is automatically verified and yields ground-truth **rewards** for the task we are attempting to accomplish. *In the real world, none these are true by default!* Instead, we must implement them ourselves, as well as any task-specific safety features that are necessary to avoid accidentally interfering with humans or damaging the robot or the environment.

## 4 Autonomously Improving Navigation Foundation Models

We apply RL to train navigation foundation models end-to-end with the following recipe:

1. Apply offline RL to a large, multi-embodiment dataset of diverse navigation data, to get a a goal-conditioned policy with broad generalization.

2. Use a navigation-specific "autonomy supervisor" to provide tasks, resets, and guardrails for exploratory behavior during deployment.

3. Learn online with a sample-efficient actor-critic algorithm to fine-tune the model with autonomous data from deployment to enable in-the-wild improvement.

See Fig. 3 for a detailed diagram explaining how all of the following components interact. We provide an implementation of LiReN and our autonomy supervisor, along with sample data collected during online RL deployment, in our supplementary material.

### 4.1 Training Setup

We perform both (offline) pretraining and (online) fine-tuning with reinforcement learning, meaning that we can use the same RL objective throughout. We train using CQL [33] both online and offline; all relevant hyperparameters are detailed in App. A.

**Reward objective.** We penalize the *final* distance to the goal, at the end of a trajectory, with terminal value function $V(s) = -\|s - g\|$. This could be accomplished with a single sparse reward for goal-reaching, but we equivalently express it with a dense reward:

$$V(s) = -\|s - g\| = \sum_t \left( \|s_t - g\| - \|s_{t+1} - g\| \right) \approx \sum_t \frac{\vec{v} \cdot \vec{sg}}{\|s - g\|}, \tag{3}$$

where $v$ is the velocity vector and $\vec{sg}$ is the vector in the direction of the goal. This step reward function $r(s, a, s', g) = \|s - g\| - \|s' - g\|$ corresponds to the *speed-made-good* [10], maximizing (long-term) speed towards the objective. Collisions are penalized with a negative reward $r_{\text{crash}}$.

**Goal relabeling.** In offline data, goals are selected from both future observations in the same trajectory (positives, sampled from an exponential distribution) and observations from other trajectories (negatives, sampled uniformly from other trajectories from the same robot embodiment), as in hindsight experience replay [32] or goal-conditioned supervised learning [34]. During online training, we keep the initially commanded ("on-policy") goals with 50% probability and otherwise relabel with positive goals. When relabeling, rewards are recomputed with the new goal.

**Privileged critic information.** In actor-critic reinforcement learning, it is crucial to have accurate critic values in order to learn a high-quality policy. We again make particular use of the problem structure of goal-conditioned navigation. While at evaluation time the policy does not require a ground-truth goal displacement vector, we *can* use this vector when training on a completed trajectory by integrating locally-accurate odometry measurements. We take advantage of this property by supplying the critic network (but not the actor) with relative goal vectors $\vec{sg}$.

**Dataset.** Single-embodiment robot models train on a relatively small dataset, collected specifically for the downstream task and environment. However, including diverse data tends to improve generalization and robustness [21]. Following ViNT [7], we train on the GNM dataset [27], a compilation of several robot navigation datasets in distinct environments and embodiments. This allows LiReN to benefit from diverse data on which to train its initial policy and learn generalizable features.

In the online phase, we continue to train on the offline dataset but draw half of each batch from the online replay buffer. This ensures that we do not experience catastrophic forgetting of behaviors learned from the original dataset, and generally stabilizes training.

## 4.2 Deployment and Autonomous Fine-tuning

We further fine-tune LiReN "on-the-job" during online deployment. In this phase, we equip LiReN with several components to propose goals, avoid getting stuck, and ensure stable operation.

**Proposing and evaluating goals.** To propose goals (represented by images) and evaluate when they are completed, we make use of the spatial structure of navigation. Prior to deployment we map the area to create a *graph* where nodes represent states (image observations) and edges represent environment transitions. During deployment, when a goal is reached, a future goal image can be selected randomly from its successors in the graph. We typically select goals 10-20 meters into the future according to an exponential distribution. We keep a static goal grpah, though if desired the graph can be updated online as in Shah et al. [11].

**Recovery behavior.** If the robot becomes stuck in a small area without exploring the rest of the environment, continuing to train on the new data will not increase overall performance. We avoid getting caught in short "goal loops" by biasing the goal graph's construction away from short cycles. Additionally, to avoid the case when the robot is unable to reach *any* goal because it is stuck or in collision with an object, we inject pseudo-resets [10, 24] to perturb the state when the robot collides with an object (detected by the robot's bumper). When it does not reach its goal for a set period $T$, it is considered "stuck" and a new goal is selected from the graph.

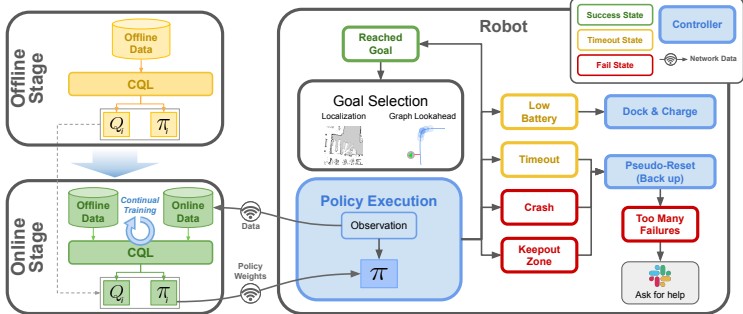

**Figure 3: System diagram of LiReN.** We use an offline dataset to train a generalist policy and $Q$-function on a broad array of navigation data. During fine-tuning the robot continually accepts new network weights from the training process and sends back data, while deciding whether to run the policy, recovery behavior, or docking fallbacks depending on the state of the robot. Goal selection is accomplished by sampling nearby goals from a pre-recorded goal graph.

**Keepout zones.** While exploring the state space, there are some zones in the environment the robot cannot enter for a variety of reasons. To protect privacy, ensure robot safety, and maintain reasonable localization, we mark areas such as bathrooms. When the robot detects that it is in a keepout zone, it triggers a pseudo-reset.

**Calling for help.** If the robot is unable to exit its current state (as measured by repeatedly becoming stuck or crashing at least ten times in a row) it will "call for help" by messaging a human over the internet. While this capability is necessary for truly lifelong learning, we find that it is triggered extraordinarily rarely (e.g. it is *physically* stuck); typically the combination of pseudo-resets and reward-guided exploration is enough for the robot to escape any local minimum.

**Battery charge monitoring.** To enable indefinite operation for lifelong learning, we continuously monitor the battery's charge. We place charging stations in the deployment environment; if the battery level drops below a set threshold, it will dock for charging the next time it passes a station.

**Robot hardware.** We use an iRobot Create 3 with an inexpensive fisheye camera. Inference is performed with a Jetson Orin NX at 3Hz. We also equip our robot with a LiDAR sensor and a map of the environment for localization, though we emphasize that this is an artifact of our goal selection mechanism, is not used by the policy, and could be avoided with a different choice of training task.

**Asynchronous training system design.** During fine-tuning, training is handled asynchronously by a server (a single Google TPUv4 node) which receives data from the robot in real time and continuously updates the robot's model parameters over the network. The training server is robust to unstable network conditions, allowing gracefully recovery from disconnects.

## 5   Experiments

We design our experiments to answer the following questions:

**Q1.** Does our combination of offline pretraining with autonomous online fine-tuning yield significant improvements in quantitative and qualitative navigation behavior?

**Q2.** Is offline RL necessary? That is, can we achieve similar autonomous fine-tuning with goal-conditioned behavior cloning (GCBC) or via online RL from scratch?

**Q3.** How performant is the offline policy learned by offline RL, in comparison with existing state-of-the-art foundation models for navigation?

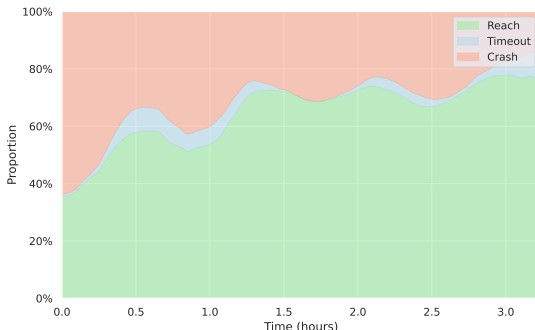

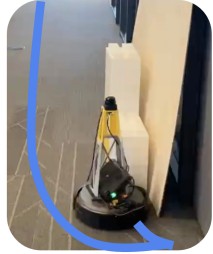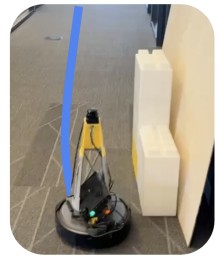

**Figure 4: Lifelong learning with LiReN.** Our model continues to train successfully without human interventions or demonstrations, improving from less than 40% initial goal-reaching performance to 75% after a few hours of autonomous online training.

**Figure 5: Qualitative improvement during online fine-tuning.** The pre-trained model cannot differentiate the out-of-distribution white obstacle from the walls. Encountering the obstacle later during fine-tuning, LiReN has learned from its experiences and successfully avoids it.

## 5.1 Evaluating Online Learning

We deploy LiReN with our autonomous improvement framework in a real-world office building. Operating in this regime, the robot must contend with diverse scenes, significant changes in lighting, and people working in the space and reconfiguring it as one might expect in an office building. We track two key metrics: **(i) success rate**, the percentage of goals successfully reached as a moving average over a window of $T = 12$ minutes (corresponding to roughly the time taken to traverse the whole environment) and **(ii) coverage**, the percentage of free space that has been explored: $\frac{1}{|\mathcal{F}|}|\cup_t \{p \in \mathcal{F} : \|p_t - p\| \leq r\}|$, where $\mathcal{F}$ is free space. It is important to consider both of these metrics, as a policy that cannot escape an "easy" region may have high success rate but low coverage.

Using our fine-tuning framework leads to significant quantitative improvements in policy performance during online deployment: LiReN trained online significantly outperforms the base policy. Figure 4 shows a steady increase in policy performance throughout the course of training.

We also see specific qualitative improvements during training. Figure 5 shows one such challenging case: the white obstacle is difficult to distinguish from the white walls, and the base model fails to navigate the situation when it is encountered for the first time. However, after a period of online fine-tuning, LiReN successfully recognizes and avoids the obstacle.

## 5.2 Should we really fine-tune with RL?

**Why not GCBC?** Interestingly, this improvement is not mirrored by attempts to fine-tune GCBC models [34]. In this setting, we train a policy (via BC) to reach relabeled goals [32], chosen from future observations from the same trajectory. We find that performing this "iterated GCBC", in which the policy is trained to match its own successful output, causes the policy's performance to degrade over time as shown in Fig. 7. We hypothesize that this is due in part to distribution shift between

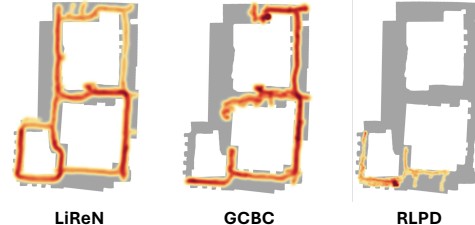

**Figure 6: Locations visited during online fine-tuning**. LiReN is explores the entire environment successfully. Iterated GCBC also exhibits reasonable coverage. Without a generalist policy initialization, RLPD becomes stuck in the initial region and cannot reach long-horizon goals.

the goals on which the policy is trained (from future states) and the "real" goals provided to the policy, which come from the fixed goal graph collected in different circumstances (e.g. lighting).

**Why not online RL?** We also consider the other end of the spectrum: applying online RL with a random initial policy. In this setting, we initialize a random policy and apply RL with Prior Data [31]

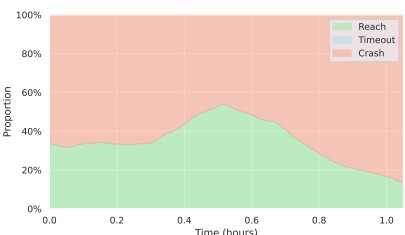

| Method | Time | Success % | Coverage % |
|---|---|---|---|
| LiReN | 0h | 32% | – |
| GCBC [34] | 1h | 11% | 63% |
| RLPD [31] | 1h | 46% | 27% |
| LiReN | 1h | **61%** | **95%** |
| LiReN | 3h | **75%** | **100%** |

**Table 1: Online Fine-Tuning.** LiReN can be fine-tuned with autonomous data to great effect. Within a few hours of online training to adapt to a new environment, LiReN's performance on goal-reaching tasks doubles that of the original policy.

**Figure 7: GCBC fine-tuning.** While GCBC gives reasonable initial performance, it cannot improve over time and performance eventually collapses. Metrics averaged over 12 minutes.

(RLPD), which trains a SAC-like policy on a mixture of offline and online data. While RLPD learns surprisingly good collision avoidance behavior in a small section of the environment, we find that it strongly overfits to the initial location and is unable to generalize. This is reflected in the coverage metrics shown in Table 1 and in the maps visualized in Figure 6.

### 5.3 Measuring Offline Performance

We adopt a methodology similar to prior work [27, 7, 35] to compare the quality of the offline policy with existing models along small, controlled sequences of goals (rather than the open-world performance reflected by the first row of Tab. 1). For each environment in which we evaluate offline performance, we collect a fixed se-

| Method | Indoor | | | Outdoor | Avg. |
|---|---|---|---|---|---|
| | Env. A | B | C | D | |
| LiReN | 34% | **97%** | **98%** | **93%** | **81%** |
| LiReN (fine-tuned) | **78%** | 78% | **91%** | **93%** | **84%** |
| GCBC | 68% | 61% | 30% | 20% | 46% |
| ViNT [7] | **71%** | 38% | 23% | 52% | 46% |

**Table 2: Offline Performance.** LiReN delivers strong zero-shot visual navigation performance, comparable to state-of-the-art models [7].

quence of 4-5 short term goals, following evaluations in Shah et al. [27]. We measure each model's progress from a consistent starting point, computed as a percent of the total distance traveled and averaged over ten trials. In addition to indoor experiments, we also provide experiments using the Clearpath Jackal operating in an outdoor setting, demonstrating LiReN's cross-embodiment capabilities. We include diagrams of each environment in App. B.

While improving pure offline performance is not the focus of this work, we nevertheless find that LiReN achieves strong empirical performance with offline RL in comparison to state-of-the-art models as highlighted in Table 2.

## 6 Discussion and Limitations

We presented LiReN, a navigation foundation models that can learn autonomously in a lifelong setting while deployed. By pre-training with offline RL on diverse navigation data, we get a visual navigation foundation model comparable to state-of-the-art models. By pairing LiReN with an autonomous improvement framework for navigation including goal selection and verification, recovery/reset procedures, and guardrails to ensure continuous operation, we can fine-tune our foundation model to continuously adapt to the target environment, learning to handle changes in the downstream environment without requiring additional human-collected data. We hope that LiReN will lay the groundwork for further study into methods for autonomously fine-tuning robot foundation models.

**Limitations and future work:** our current autonomous improvement framework does have several limits: our goal selection procedure currently requires a notion of localization to find and verify nearby locations on the goal graph, and we currently only support a single fine-tuning task. Additionally, while the robot we used for deployment and online improvement was not itself in the dataset, we did not explore concurrently fine-tuning RL models with multiple new embodiments at once; we expect that this might raise its own challenges.

## Acknowledgments

This research was partly supported by the NSF under IIS-2150826, ARL DCIST CRA W911NF-17-2-0181, and ONR N00014-22-1-2773. The authors would like to thank Catherine Glossop, Dhruv Shah, Zhiyuan Zhou and Pranav Atreya for helpful discussions early in the project. We would also like to thank Oier Mees for feedback on an early draft of the manuscript and its figures.

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

# Appendix A   Network Architecture and Hyperparameters

**Model Architecture**

| | | |
|---|---|---|
| **Encoder** | Structure | ResNet18 |
| | Pooling | spatial softmax, 8 blocks |
| **Critic** | Structure | MLP+repeat action conditioning [21] |
| | Normalization | LayerNorm |
| | Hidden dims | (256, 256) |
| **Actor** | Structure | MLP |
| | Hidden dims | (256, 256) |

**Training**

| | | |
|---|---|---|
| **CQL** | Discount | 0.97 |
| | Target entropy | -1.0 |
| | Temperature init | 0.1 |
| | DR3 regularization | 1e-3 |
| | Alpha | 0.3 |
| **Optimizer** | Batch size | 256 |
| | Optimizer type | adamw |
| | Weight decay | 1e-4 |
| | Agent LR | 0.0001 |
| | Critic LR | 0.0003 |
| | Temp LR | 0.0003 |

**Data**

| | | |
|---|---|---|
| **Image transforms** | Image size | 64x64 |
| | Augmentations | random brightness/contrast/hue, random flip, random crop |
| **Offline dataset goal sampling** | Positive probability | 0.75 |
| | Negative probability | 0.25 |
| | Sampling distribution | Exponential(20) |
| **Online dataset goal sampling** | Original goal probability | 0.5 |
| | Positive probability | 0.375 |
| | Negative probability | 0.125 |
| | Data before training | 1500 |

**Autonomy**

| | | |
|---|---|---|
| **Goals** | Goal reach distance | 0.75 meters |
| | Goal reach angle error | 45 degrees |
| | Sampling distribution | Exponential(5) |
| **Error handling** | Max failures in a row | 10 |
| | Low battery threshold | 25% |

**Table 3:** Model Architecture and Training Parameters

## Appendix B    Offline Testing Environment Layouts

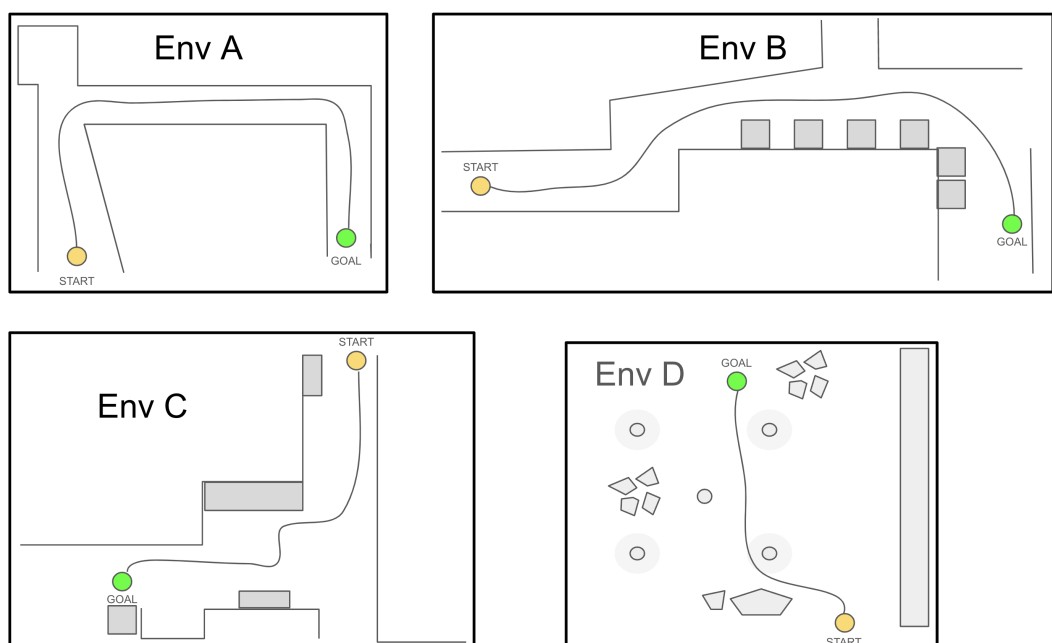

**Figure 8:** Small environments for evaluation of offline policies. Goal trajectories are depicted, along which 4-5 goal waypoints are placed. When a goal is achieved in the offline evaluation setting, we advance the goal deterministically to the next waypoint.

