# OpenReview forum: "Lifelong Autonomous Improvement of Navigation Foundation Models in the Wild"
_robot-learning.org/CoRL/2024/Conference — CoRL 2024_

### Official Review · Reviewer_p4fP · 2024-07-15
**Good contribution to lifelong robot learning, but somewhat "oversold"**

**Originality:** 3
**Technical Quality:** 4
**Clarity Of Presentation:** 3
**Potential Impact:** 3
**Recommendation:** 4
**Confidence:** 4

**Review:**

######## Update after rebuttal ########
Per the discussion with the authors below, I am updating my recommendation to "Strong accept".

######## Strengths ########

1. This paper is, to my knowledge, the first plausible instantiation of a lifelong learning method on a physical robot.
    - Not many existing works have attempted to learn continually over long deployments in robots. This is the most successful version of such attempts that I am aware of. I do encourage the authors to contextualize their work with respect to such existing attempts (e.g., [1], [2])
    - Compared to those existing works, this submission studies a setting that is both autonomous (there is no human to provide ground-truth rewards or reset the environment at will) and online (the agent learns during deployment while it attempts to solve the tasks that it is intended to solve)
2. The primary technical strength of the work lies in the design of a pipeline that enables _fully autonomous_ learning on a physical robot in a natural environment.
    - What makes this work promising toward true lifelong robot learning is that the authors carefully design a pipeline/framework for completely autonomous learning/evaluation. The framework proposes goals, constructs rewards measured from the agent's own sensors, incorporates resetting, battery charging, error recovery, and (as a failsafe) "asking for help."
    - The closest design I had seen to this was in CMU's Cobots [3], which the authors should also likely cite. Critically, [3] was not a _learning_ robot, but simply a long-term deployed robot. The current submission steps this up by introducing all the relevant mechanisms to use during online learning.
3. The experiments are well designed and demonstrate the superiority of the proposed approach over well-chosen baselines.
    - The experiments focus on showing that pure offline training is not enough, that methods pretrained via behavior cloning instead of offline RL underperform, and that pure online training is comparatively slow.
    - The authors include two metrics: success rate over the proposed goals, and space coverage. This is well appreciated, because success rate on its own could be misleading (and, as a reader, I likely would have missed it if not because the authors included the space coverage metric). The key point here is that since the pipeline itself proposes local goals, if the agent fails at goals "on the border" of the local region, then it might never be able explore beyond, while still being able to achieve the "internal" goals within the explored region. It might be worth stating this more plainly up front.
    - Overall, the results are strong, though it seems necessary to include additional details (as described below)

######## Weaknesses ########

1. The authors claim that their approach improves foundation models via lifelong learning, which is much more general than what their method is designed for (or demonstrated to achieve).
    - This is my main concern: I believe that the authors should rephrase their contribution away from "improving robot foundation models."
    - The purpose of a foundation model is to be a general-purpose model to work across a variety of robot morphologies, environments, and objectives. Improving a foundation model would require addressing these dimensions both in the method and in the evaluation, neither of which are done in this submission.
    - The authors vaguely hint at this in the final section: "we do not support fully multi-task foundation models" and "we did not explore concurrently fine-tuning RL models with multiple new embodiments at once without experiencing forgetting." But this is insufficient: I encourage the authors to rephrase the contributions of their work to more precisely characterize them.
    - In particular, the proposed method permits "adapting", "finetuning", or "specializing" a foundation model to a particular embodiment-environment pair throughout a lifelong deployment. This is already, in my opinion, a strong contribution.
2. The general idea of "offline RL -> online finetuning" is not novel (though the details of how it is deployed might be).
    - This somewhat limits the novelty of the contribution, though there is still sufficient technical novelty for publication.
    - While the authors cite [22] as an existing approach that pretrains a model via offline RL and adapts it online on a physical robot, I think it might be worth directly stating that the whole motivation for the existence of the offline RL field is precisely to train models on large-scale data. While today the "popular" technique is to use behavior cloning, large-scale pretraining has long been proposed as a key setting for offline RL. I encourage the authors to cite relevant literature in this space and contextualize their contribution accordingly.
3. The method description is insufficient to understand the goal relabeling scheme, which I believe to be a key component of the proposed method.
    - The authors mention a couple of times that they relabel states/goals, but they never explicitly describe this.
    - "we only relabel 20%...", "we _do_ have access to this information when relabeling states..." --> this isn't made explicitly clear, which is why I'm confused: do the authors use something like hindsight experience replay [4] to make some of the online data look like positive data?
    - I encourage the authors to include an additional paragraph in Sec 4 that states what the relabeling process is.
4. There are several missing details to fully understand the experimental settings.
    - "Interestingly, this improvement is not mirrored by attempts to fine-tune goal-conditioned behavior cloning models" --> in prior work or where? Or do the authors mean the results in Fig 7, which is never referenced? This sentence isn't clear.
        - Assuming that this means Fig 7, how is the behavior cloning model finetuned without demos? Do the authors use the attained goals as the true goals for training?
    - "pure online RL without a base policy (but taking into account prior data)" --> how is prior data "taken into account"?
    - The whole setup of Sec 5.3 is quite unclear -- is "offline performance" the performance of the pretrained models deployed on the robot online without any finetuning? Why is the performance of LiReN different from that in Table 1 at 0h?

[1] Sutton et al. "Horde: A scalable real-time architecture for learning knowledge from unsupervised sensorimotor interaction." AAMAS 2011.

[2] Isele et al. "Lifelong learning for disturbance rejection on mobile robots." IROS 2016.

[3] Biswas and Veloso. "Localization and Navigation of the CoBots Over Long-term Deployments." IJRR 2013.

[4] Andrychowicz et al. "Hindsight Experience Replay." NIPS 2017.

**Quality Of The Limitations Section:**

3

**Questions For Rebuttal:**

Summarizing the points above, here are the main questions/suggestions I have for the authors:
- Please include the relevant missing references (and discussions about their connection to this work)
- Please scope the contribution appropriately in terms of specializing or finetuning (instead of the more general "improving")
- What is the exact relabeling process used by the authors? Is this novel or taken from existing work?
- Please answer my questions above about the experimental setting

######## Additional feedback ########

The following points are provided as feedback to hopefully help better shape the submitted manuscript, but did not impact my recommendation in a major way.

Abstract
- It is unclear at this point what "a complete system" is

Intro
- Either define "behavior cloning (BC)" and use "BC" thereafter, or drop "BC" and stick to "behavior cloning"
- "a critic $Q(s,a)$, which requires different feature-learning capabilities than behavior cloning" --> could the authors argue this point? Is there existing literature that studies whether this is true, or is there some obvious reason why features learned for BC couldn't be used for Q?

Sec 2
- This is the second time the authors mention a "supervisory framework to enable autonomous learning", but this doesn't really mean much on its own. It might be worth to include ~1 sentence in the intro that broadly describes what this entails.

Sec 3
- This section seems quite precise, which is nice.

Sec 4
- I suggest separating Sec 4 into two sections: one focusing on the non-learning aspects that are necessary for safe long-term deployment (roughly p to line 178) and the learning aspects (from line 179 on, plus maybe the goal proposal)
- There's a sense in which the goal proposal really is just a "trick" to enable evaluating these ideas. In the real world, we would want the robot to learn from actual goals that users intend for it (or goals that it sets for itself with some real higher-level objective, like surveillance). It might be worth discussing this point. More generally, this section would benefit from additional clarity around which aspects are necessary for training, for longterm autonomy, or for evaluation.
- I didn't really understand the "privileged critic information" paragraph, but I think this relates to the relabeling.
- "we emphasize that this is not necessary for downstream deployment" --> this is a really odd comment in a lifelong deployment setup. I think the authors mean that the LiDAR is only used to set the goal, which is only an artifact of the evaluation setting and not the actual learning setting.

Sec 5
- I like the design of fig 4, like a continuous stacked bar chart
- What is the cause for the oscillating pattern in the performance improvement?
- The fact that RLPD obtains 46% success rate but only covers 27% of the space suggests that success rate over the distribution of proposed goals is not on its own a great metric of performance. But including the results on coverage addresses this.

**Robotics Focus:**

4

**Summary Of Paper:**

The submission proposes a pipeline for lifelong learning for robot navigation by starting from a pretrained model. In particular, the method uses large-scale data to pretrain a policy and Q-function via offline RL (using conservative Q-learning; CQL [28]0), and then uses the learned models as a starting point for adapting to a set of real-world navigation tasks. The pipeline is fully autonomous: it provides goals for the robot and recovery mechanisms to avoid the need for human interventions except in the most extreme cases where the robot is physically stuck. The experiments demonstrate improvements over existing methods, which also serve as ablation studies for the proposed approach.

**Summary Of Recommendation:**

I lean toward recommending acceptance for the submission. While in terms of learning the contributions are not significant, they are in terms of proposing a complete pipeline that enables learning with existing methods within a physical platform. The authors should primarily address two concerns: delimit the scope of their contribution more precisely, and describe more clearly the details of their goal relabeling scheme.

---

### Official Review · Reviewer_4zSa · 2024-07-18
**Relevant topic and nice idea**

**Originality:** 3
**Technical Quality:** 3
**Clarity Of Presentation:** 2
**Potential Impact:** 3
**Recommendation:** 4
**Confidence:** 4

**Review:**

**Quality, clarity, originality, and significance**:
The paper is easy to read and well-organized, and I was able to follow the arguments made by the authors mostly without problems. This paper addresses an important topic in robotics that is often overlooked: how a robot can continue to improve its capabilities by learning from its own experience after deployment. The authors discuss the motivation behind this work clearly and present relevant real-world experiments. In my view, combining offline pretraining with online adaptation and the use of the "autonomy supervisor" to minimize human involvement are original ideas that can also potentially be suitable for robot modalities beyond navigation.  Though the overall quality of the paper is good, certain aspects need to be improved (see weaknesses, and questions below).

**Strengths**:
1. The general framework and the idea behind this work are appealing. Combining pretraining on general data with offline RL and adaptation in the real world with sample-efficient online RL is a nice idea that can be relevant to other areas of robotics as well.

2. Reliance on real-world experiments and data.

3. In general, the paper is well-written.

**Weaknesses**:
1. Insufficient details for results (please see questions for rebuttal).

2. A single architectural diagram to describe the framework would help the reader quickly grasp the different components of the proposed framework. This would be easier than deciphering separate figures (e.g. figs. 1,2,3). Also, the textual description of the overall system would benefit.

3. A concise description of LiReN's training algorithm, i.e. in the form of a pseudocode listing, is missing (for both offline and online RL). It may help to improve the clarity of the paper.

4. I appreciate that the authors have provided code and data, but the supplementary material should also describe network architectures, model sizes, hyperparameters, etc. in text/tabular form.

5. Multiple equations are not numbered (e.g. on pages 3, 5).

**Quality Of The Limitations Section:**

3

**Questions For Rebuttal:**

1. How many independent trials were used to calculate the results reported in Figs. 4, 7, and tables 1 and 2?

2. What is the effect of the size of the dataset used for pretraining via offline RL? How much data is sufficient?

3. Line 119: "... some additional information about $s_t$ becomes knowable later" - Why does this information become available later? When does it become available?

4. Line 116: The authors write "In LiReN’s implementation, we use a combination of all three of these techniques ..." but the previous line mentions 4 techniques: clipped double-Q, ensembling, entropy regularization, and critic normalization.

5. How is a crash detected?

6. Why do you not consider any penalty in the reward formulation (e.g. a penalty for crashing)?

7. Concerning Fig. 7, why does the performance of GCBC collapse after improving initially?

8. Do you map the environment manually before deploying the robot? Please mention this in the paper.

9. In Table 1, how do you calculate the coverage percentage?

10. What are the differences between "Env A", "Env B" and "Env C" in Table 2?

11. Is it a good idea to perform online training continuously? There is a difference between finetuning and lifelong/continual learning. For example, suppose the robot trains during the day (with some natural light) and then continues to adapt (i.e. finetune its parameters) to artificial light at night. Does it need to readapt to natural light the next day? As there is no mechanism to prevent forgetting, I would assume the answer to this question is yes. This is possibly something to look into in the future but I would be interested to know the authors' view on this.

12. *Minor*: Line 164: Is this a typo? "... it triggers a minimal reset which backs away it is back ..."

**Robotics Focus:**

4

**Summary Of Paper:**

This paper proposes a method to continuously and autonomously improve the capability of navigation foundation models after deployment in a real-world environment. It leverages offline pretraining on diverse navigation data and later adapts the policy on the real robot via sample-efficient online RL.

**Summary Of Recommendation:**

I like the proposed idea and feel that the experimental results mostly support the claims made by the authors. At a high level, the framework proposed in the paper is quite clear, but it is somewhat difficult to completely grasp all the details because of the absence of a concise description of all the training procedure. Some aspects of the results also need further clarification (e.g. how many independent trials were conducted).

---

### Official Review · Reviewer_BE4N · 2024-07-20
**Good experiment, but writing is confusing.**

**Originality:** 3
**Technical Quality:** 3
**Clarity Of Presentation:** 2
**Potential Impact:** 3
**Recommendation:** 3
**Confidence:** 4

**Review:**

This paper discusses an interesting problem: how to improve a navigation foundation model during deployment. This approach can serve as a helpful recipe for future research on navigation foundation models.

I find the experiment convincing, as it clearly demonstrates the impact of online RL for improvement and the performance of offline RL pre-training.

However, there are some issues that need to be addressed:

1. Since you are discussing a "Navigation Foundation Model," it is important to show that your method can be deployed on different navigation robots, similar to what Vint is doing.

2. Sections 3 and 4 are somewhat confusing. Your method involves two stages: an offline RL pre-training stage and an online RL fine-tuning stage, conducted in sequence—first offline RL, then online RL. Section 3 gives the impression that you are discussing online RL first, then switching to offline RL, and then back to the online part. In Section 4, it is unclear whether the methods are used for online RL, offline RL, or both. I suggest organizing these two steps (offline-RL, online-RL) in a sequential order and explaining exactly what you did during each step. The offline RL pre-training is a novel part of this paper and should be explained thoroughly.

**Quality Of The Limitations Section:**

3

**Questions For Rebuttal:**

1. Since you are discussing a "Navigation Foundation Model," it is important to show that your method can be deployed on different navigation robots, similar to what Vint is doing.

2. I want to see the offline-RL and online-RL parts more clearly explained. Specifically, the methods you mentioned in Section 4—during which stage are they used?

3. I also have a deployment question. From Figure 4, we can see that the performance of navigation improves over time. Does this mean that you are training on the fly, or are you collecting data for 10 minutes and then updating the policy? Perhaps you can provide more details about the deployment process in the appendix.

**Robotics Focus:**

4

**Summary Of Paper:**

This paper presents a new approach for a navigation foundation model. Offline RL is used to fine-tune a general foundation model, followed by online RL for further improvement. Real-world experiment results demonstrate the effectiveness of the proposed module.

**Summary Of Recommendation:**

I like the idea and the experiment of this paper. But I do think the writing for the section 3 and 4 needs to be improved before acception.

---

### Author Rebuttal · Authors · 2024-08-10

We thank the reviewers for the detailed suggestions. While all reviewers appreciated the importance of fine-tuning generalist robot policies with online RL, there were some concerns about (1) whether LiReN is a “foundation model” and (2) the clarity of writing. **We have uploaded a revised manuscript with major changes or new additions highlighted in green.**

We address all points individually under the main reviews, but we highlight a few major improvements to the submission in response to the points raised:

1. **New Experiments.** Several reviewers pointed out that the term “foundation model” requires evidence that the policy works on multiple embodiments. In support of our claims that LiReN is in fact a foundation model we introduce additional experiments in which we run the LiReN policy on a significantly different embodiment: the Clearpath Jackal, a larger outdoor robot. LiReN demonstrates strong performance in this setting:

| Method | Env. D |
|---|---|
| GCBC | 20% |
| ViNT | 52% |
| LiReN | 93% |

We also test unconditional collision avoidance behavior using a procedure outlined in ViNT in which a random goal image is sampled to simulate unconditional exploration. To answer questions of whether the fine-tuning procedure outlined induces forgetting, we also show results from the policy **after fine-tuning on the indoor task originally presented**, to verify that outdoor performance does not degrade after fine-tuning (see attached videos):

| Method | Exploration time before crash |
|---|---|
| GCBC | 0:23 |
| LiReN (base) | 2:37 |
| LiReN (finetuned indoors) | 3:24 |

These results demonstrate that LiReN is capable of operating on multiple navigation embodiments, and that fine-tuning on one embodiment does not significantly degrade performance on another.

2. **Writing and Organization.** We improved clarity throughout. In particular, we reorganized Section 4 to highlight which components are used both offline and online and which are only used in the online phase. We also clarify that the novelty of our method is the framework enabling a successful fine-tuning system rather than the idea of offline -> online RL itself.
3. **Appendix.** We included more details about how the components of our system interact, as well as an appendix with detailed system diagrams and hyperparameters.

We hope the reviewers find these improvements convincing. We are happy to clarify any questions that would help the reviewers improve their recommendations.

---

### Decision · Program_Chairs · 2024-09-04

**Decision:**

Accept

**Comment:**

The paper presents an online learning approach for a foundation navigation model that can be refined as the robot collects more data.

Strengths:
- Interesting idea and an effective algorithm
- Convincing experiments that clearly demonstrate the impact of online RL for improvement the performance of previously trained models

Weaknesses:
- More details needed for the results
- Clarity on the method can be improved with pseudo-code or a diagram
- Novelty needs to be better described as the general idea of offline RL followed by online finetuning has been proposed before
- Clarify on how the method improve "foundation models" in all its generality

The reviewers provided valuable suggestions to improve the paper. I encourage the authors to work with the reviewers in addressing their concerns.
===============================
Pos-rebuttal update:

The authors have addressed the concerns reported and the reviewers now agree that the paper should be accepted.